# $\ell_\infty$-Bounds of the MLE in the BTL Model under General Comparison Graphs

**Wanshan Li**[1]     **Shamindra Shrotriya**[1]     **Alessandro Rinaldo**[1]

[1]Department of Statistics & Data Science , Carnegie Mellon University , Pittsburgh, Pennsylvania, USA

## Abstract

The Bradley-Terry-Luce (BTL) model is a popular statistical approach for estimating the global ranking of a collection of items using pairwise comparisons. To ensure accurate ranking, it is essential to obtain precise estimates of the model parameters in the $\ell_\infty$-loss. The difficulty of this task depends crucially on the topology of the pairwise comparison graph over the given items. However, beyond very few well-studied cases, such as the complete and Erdös-Rényi comparison graphs, little is known about the performance of the maximum likelihood estimator (MLE) of the BTL model parameters in the $\ell_\infty$-loss under more general graph topologies. In this paper, we derive novel, general upper bounds on the $\ell_\infty$ estimation error of the BTL MLE that depend explicitly on the algebraic connectivity of the comparison graph, the maximal performance gap across items and the sample complexity. We demonstrate that the derived bounds perform well and in some cases are sharper compared to known results obtained using different loss functions and more restricted assumptions and graph topologies. We carefully compare our results to Yan et al. (2012), which is closest in spirit to our work. We further provide minimax lower bounds under $\ell_\infty$-error that nearly match the upper bounds over a class of sufficiently regular graph topologies. Finally, we study the implications of our $\ell_\infty$-bounds for efficient (offline) tournament design. We illustrate and discuss our findings through various examples and simulations.

## 1 INTRODUCTION

Simultaneous or 'global' ranking of a set of items is a practical problem that arises naturally in a variety of domains.

For example, one may wish to ascertain a 'best player' or 'best team' in a given sports league. Designing a principled statistical approach to global ranking of items is challenging due to data limitations and complex domain-specific relationships between the underlying items to be ranked.

A popular and practicable solution to estimating global ranking is to utilize pairwise comparison information across the items to be ranked, which is easily accessible across many application domains. The BTL model (Bradley and Terry, 1952; Luce, 1959) is a popular statistical model for pairwise comparison data. A similar model was also originally studied in Zermelo (1929). The continued practical and theoretical interest in the BTL model stems from its relatively simple parametric form which provides a good balance between interpretability and tractability for theoretical analysis. The BTL model is domain-agnostic, making it an ideal benchmarking tool across a variety of ranking applications e.g. sports analytics (Fahrmeir and Tutz, 1994; Masarotto and Varin, 2012; Cattelan et al., 2013), and bibliometrics (Stigler, 1994; Varin et al., 2016).

Formally, we can describe the BTL model as follows. Suppose that we have $n$ distinct items, each with a (fixed but unobserved) positive strength or preference score $w_i^*$, $i \in [n]$, quantifying item $i$'s propensity to beat other items in pairwise comparisons. The BTL model assumes that the comparisons between different pairs are independent and the outcomes of comparisons between any given pair, say item $i$ and item $j$, are i.i.d. Bernoulli random variables, with *winning probability* $p_{ij}$, defined as

$$p_{ij} := \mathbf{Pr}\,(i \text{ beats } j) := \frac{w_i^*}{w_i^* + w_j^*}, \ \forall\, i, j \in [n]. \quad (1)$$

A common reparametrization is to set, for each $i$, $w_i^* = \exp(\theta_i^*)$, where $\boldsymbol{\theta}^* := (\theta_1^*, \ldots, \theta_n^*)^\top \in \mathbb{R}^n$. By convention, we assume that $\sum_{i \in [n]} \theta_i^* = 0$ for parameter identifiability.

From a theoretical perspective, much attention in the BTL literature has been paid to two popular estimators, namely the maximum likelihood estimator (MLE) and the spectral

*Accepted for the 38th Conference on Uncertainty in Artificial Intelligence* (UAI 2022).

method (Jain et al., 2020). Recently, Chen et al. (2020) show that the MLE attains a sharper minimax rate of the Hamming top-$k$ loss compared to the spectral method. In this paper, we thus focus on the MLE, which we formally define later in Section 2.

### General pairwise comparison graphs

Given $n$ items to be compared, the pairwise comparison scheme among them can be expressed through an undirected simple graph $\mathcal{G}(V, E)$, where the vertex set $V := [n]$ and the edge set $E := \{(i, j) : i \text{ and } j \text{ are compared }\}$ is determined by the comparison scheme. Correspondingly, if we define the directed edge set as $E_d := \{(i, j, k) : (i \text{ beats } j) \ k \text{ times}\}$, then the induced directed simple graph $\mathcal{G}(V, E_d)$ is called a *directed* comparison graph. It is a classical result (Ford, 1957; Simons and Yao, 1999; Hunter, 2004) that the BTL model is identifiable if and only if $\mathcal{G}(V, E)$ is connected, and the MLE of the model parameters exists and is consistent if and only if $\mathcal{G}(V, E_d)$ is strongly connected. Henceforth, *comparison graph* refers to the undirected pairwise comparison graph.

Typically one is interested in getting sharp bounds for the estimation risk, which could be based on a norm-induced metric $\|\hat{\boldsymbol{\theta}} - \boldsymbol{\theta}^*\|_p$ or a ranking metric, e.g., Kendall's tau distance (Kendall, 1938). What makes risk analysis of BTL model estimators particularly challenging is a combination of the type of estimation risk loss considered, and the assumptions on the topology of $\mathcal{G}(V, E)$.

### Core questions of interest

Among all the metrics measuring uncertainty of estimators of BTL parameters, the $\ell_\infty$-loss directly connects with ranking metrics, e.g. binary and Hamming top-$k$ (partial) ranking loss (see, e.g. Chen et al., 2019, 2020).

It is thus natural to study the MLE for the BTL parameters in the $\ell_\infty$-loss, to better understand the risk optimality of the MLE and further justify its use for practical global and partial ranking problems. In this spirit, Yan et al. (2012) focus specifically on proving $\ell_\infty$-error bounds for the BTL MLE for general comparison graphs. However, a notable limitation in their setting is that they impose a strictly dense comparison graph assumption, which may be impractical in many real world applications. This leaves a gap in the literature, summarized in the following questions:

> ***Core questions:*** *For the BTL model, how does the MLE perform with respect to the $\ell_\infty$ loss, under much weaker assumptions on the pairwise comparison graph compared to Yan et al. (2012)? That is, assuming only that the comparison graph is connected. Moreover, what are the implications of such bounds in applications?*

Providing a sharp analysis to these questions with a detailed comparison to recent theoretical results in the BTL literature motivates our work in this paper.

### Relevant and related literature

We give a brief overview of the work that addresses the challenge of comparison graph topology in ranking. When the comparison graph is a complete graph, Simons and Yao (1999) give a high-probability upper bound for the $\ell_\infty$ loss, i.e., $\|\hat{\boldsymbol{\theta}} - \boldsymbol{\theta}^*\|_\infty$ and obtain the asymptotic distribution of the MLE. In the setting where the comparison graph follows the Erdös-Rényi graph model, Chen and Suh (2015), Chen et al. (2019), Chen et al. (2020) and Han et al. (2020) derive high-probability upper bounds for the $\ell_\infty$ loss. Moreover, Chen et al. (2019) show that both MLE and spectral method are minimax optimal in terms of the binary top-$k$ ranking loss, i.e., whether the items with the highest $k$ out of $n$ preference scores are perfectly identified; Chen et al. (2020) consider a Hamming Loss for top-$k$ items and show that the MLE is minimax optimal compared to the spectral method with differences arising in constant factors.

For a broader class of comparison graphs beyond complete and Erdös-Rényi graph, researchers have studied the explicit dependence of the estimation risk on graph topology. In particular, Yan et al. (2012) give a high-probability upper bound for the $\ell_\infty$-loss for relatively dense graphs. Hajek et al. (2014); Shah et al. (2016) give a high probability upper bound for the $\ell_2$ or Euclidean loss $\|\hat{\boldsymbol{\theta}} - \boldsymbol{\theta}^*\|_2$, establish upper and lower bounds of $\mathbb{E}\|\hat{\boldsymbol{\theta}} - \boldsymbol{\theta}^*\|_2$ and show the minimax optimality of the constraint MLE across a wide range of graph topologies. Recently, Agarwal et al. (2018) give sharp upper bounds for a novel spectral method in the $\ell_1$-loss $\|\hat{\boldsymbol{\pi}} - \boldsymbol{\pi}^*\|_1$ for $\boldsymbol{\pi}^* = \mathbf{w}^*/\|\mathbf{w}^*\|_1$ instead of $\boldsymbol{\theta}^*$. Hendrickx et al. (2019, 2020) propose a weighted least square method to estimate $\mathbf{w}^*$ and prove a sharp upper bound for their estimator in $\mathbb{E}[\sin^2(\hat{\mathbf{w}}, \mathbf{w}^*)]$ or equivalently in $\mathbb{E}\|\hat{\mathbf{w}}/\|\hat{\mathbf{w}}\|_2 - \mathbf{w}^*/\|\mathbf{w}^*\|_2\|_2^2$, in the sense that this upper bound matches a instance-wise lower bound up to constant factors.

### Contributions

Our contributions in this paper are fourfold and are summarized as follows:

- **Upper bounds:** We derive a novel upper bound for the $\ell_\infty$-error of the regularized MLE in BTL model allowing for general graph topology. Our upper bounds hold under minimal assumptions on graph topologies, i.e., assuming only that the comparison graph is connected. Given such generality, we show our $\ell_\infty$ bound is tighter than existing results under a broad range of graph topologies, and works well in general. In particular, we carefully compare our work analytically and in simulation to Yan et al. (2012), which is closest in spirit to our work.

  A minor corollary of our techniques results in the state

of the art $\ell_2$-loss bounds for the Erdös-Rényi graph.

- **Lower bounds:** We derive minimax lower bounds for BTL parameter estimation in $\ell_\infty$-loss. We analyze specific graph topologies satisfying certain regularity connectivity conditions under which the BTL MLE is nearly minimax optimal.

- **Implications for tournament design:** We show that the BTL MLE in $\ell_\infty$-loss satisfies a unique subadditivity property, and how our $\ell_\infty$ bounds can exploit this property for efficient (offline) tournament design.

- **Extension to the unregularized BTL model:** We also extend our upper bounds under $\ell_\infty$-loss to the unregularized ('vanilla') BTL MLE, which is also frequently used in practice.

Due to the more complicated form of the vanilla BTL MLE upper bounds and space limitations, we present these analagous results and their proofs separately in Appendix A.7. Henceforth, MLE refers to the regularized BTL MLE unless stated otherwise. In addition to our theoretical contributions a core aspect throughout our paper is to emphasize the interpretability of our results, the associated assumptions, and implications for practical ranking tasks.

### Organization of the paper

The rest of the paper is organized as follows. In Section 2, we present our main results for the upper bound in Theorem 1 and an interpretation of the key components of the bound. In Section 3, we discuss minimax lower bounds using the $\ell_\infty$ risk loss in Theorem 5. In Section 4, we show some practical implications of our results in efficient tournament design from a ranking perspective. In Section 5, we conduct extensive numerical simulations to validate the optimality of our bounds compared to related results in the literature.

### Notation

We typically use lowercase for scalars, e.g., $(x, y, z, \ldots)$, boldface lowercase for vectors, e.g., $(\mathbf{x}, \mathbf{y}, \mathbf{z}, \ldots)$, and boldface uppercase for matrices, e.g. $(\mathbf{X}, \mathbf{Y}, \mathbf{Z}, \ldots)$. We denote the finite set $\{1, \ldots, n\}$ by $[n]$. For asymptotics, we denote $x_n \lesssim y_n$ or $x_n = O(y_n)$ and $u_n \gtrsim v_n$ or $u_n = \Omega(v_n)$ if $\forall n$, $x_n \leq c_1 y_n$ and $u_n \geq c_2 v_n$ for some constants $c_1, c_2 > 0$. We denote $\mathbf{e}_i$ as a vector whose entries are all 0 except that the $i$-th entry is 1. $a_n = o(b_n)$ means $a_n/b_n \to 0$ as $n \to \infty$ and conversely, $a_n = \omega(b_n)$ means $b_n/a_n \to 0$ as $n \to \infty$. We denote $\mathbf{1}_n \in \mathbb{R}^n$ to be a vector of ones.

## 2 UPPER BOUNDS

Recall that given $n$ items to be compared, the comparison scheme among them defines the comparison graph $\mathcal{G}(V, E)$, where $V = [n]$ and $E = \{(i, j) : i \text{ and } j \text{ are compared }\}$. We denote the corresponding adjacency matrix as $A \in \mathbb{R}^{n \times n}$, and its $(i, j)^{\text{th}}$ entry is $A_{ij} := 1\{(i, j) \in E\}$. The

associated (unnormalized) graph Laplacian is the symmetric, positive-semidefinite matrix $\mathcal{L}_{\mathbf{A}} := \mathbf{D} - \mathbf{A}$, where $\mathbf{D} = \mathrm{diag}(n_1, \ldots, n_n)$, with $n_i := \sum_{j=1}^n A_{ij}$ the degree of node $i$. It is well known that the smallest eigenvalue of $\mathcal{L}_A$ is 0 with an eigenvector $\mathbf{1}_n$. Let $\lambda_2(\mathcal{L}_{\mathbf{A}})$ be the second smallest eigenvalue of $\mathcal{L}_{\mathbf{A}}$, known as the algebraic connectivity of $\mathcal{G}$ (Das, 2004), then $\mathcal{G}$ is connected if and only if $\lambda_2(\mathcal{L}_{\mathbf{A}}) > 0$. Following the standard in the BTL literature we assume a that for each edge $(i, j)$ of the comparison graph, the corresponding items $i$ and $j$ are compared $L$ times, each leading to an independent outcome $y_{ij}^{(l)} \in \{0, 1\}$, where $l \in [L]$. If pairs are compared different number of times, we take $L$ to be the smallest number of pairwise comparisons over the edge set, as a worst-case scenario. The corresponding sample averages are denoted with $\bar{y}_{ij} = \frac{1}{L} \sum_{l=1}^L y_{ij}^{(l)}$ and are sufficient statistics for the model parameters. The $\ell_2$-regularized MLE is defined as

$$\hat{\boldsymbol{\theta}}_\rho = \underset{\mathbf{1}_n^\top \boldsymbol{\theta} = 0}{\arg\min} \, \ell_\rho(\boldsymbol{\theta}; \mathbf{y}), \; \ell_\rho(\boldsymbol{\theta}; \mathbf{y}) = \ell(\boldsymbol{\theta}; \mathbf{y}) + \frac{\rho}{2} \|\boldsymbol{\theta}\|_2^2, \; (2)$$

where $\ell(\boldsymbol{\theta}; \mathbf{y})$ is the negative log-likelihood, given by

$$\ell(\boldsymbol{\theta}; \mathbf{y}) := - \sum_{1 \leq i < j \leq n} A_{ij} \{ \bar{y}_{ij} \log \psi(\theta_i - \theta_j) \qquad (3)$$
$$+ (1 - \bar{y}_{ij}) \log[1 - \psi(\theta_i - \theta_j)]\},$$

and $t \in \mathbb{R} \mapsto \psi(t) = 1/[1 + e^{-t}]$ the sigmoid function.

Under this notational setup, we are ready to state the $\ell_\infty$ upper bound of the BTL MLE in Theorem 1.

**Theorem 1.** *Assume the BTL model with parameter $\boldsymbol{\theta}^* = (\theta_1^*, \ldots, \theta_n^*)^\top$ such that $\mathbf{1}_n^\top \boldsymbol{\theta}^* = 0$ and a comparison graph $\mathcal{G} = \mathcal{G}([n], E)$ with adjacency matrix $\mathbf{A}$, algebraic connectivity $\lambda_2(\mathcal{L}_{\mathbf{A}})$ and maximum and minimum degrees $n_{\max}$ and $n_{\min}$. Suppose that each pair of items $(i, j) \in E$ are compared $L$ times. Let $\kappa = \max_{i,j} |\theta_i^* - \theta_j^*|$ and $\kappa_E = \max_{(i,j) \in E} |\theta_i^* - \theta_j^*|$ and set $\rho \geq c_\rho \kappa^{-2} e^{-2.5\kappa_E} n^{-4} n_{\max}^{1/2}$. Assume that $\mathcal{G}$ is connected or $\lambda_2(\mathcal{L}_A) > 0$. Then with probability at least $1 - O(n^{-4})$, the regularized MLE $\hat{\boldsymbol{\theta}}_\rho$ from (2) satisfies*

$$\|\hat{\boldsymbol{\theta}}_\rho - \boldsymbol{\theta}^*\|_\infty \lesssim \frac{e^{2\kappa_E}}{\lambda_2} \frac{n_{\max}}{n_{\min}} \left( \sqrt{\frac{n+r}{L}} + \rho\kappa \sqrt{\frac{n}{n_{\max}}} \right)$$
$$+ \frac{e^{\kappa_E}}{\lambda_2} \sqrt{\frac{n_{\max}(\log n + r)}{L}}, \qquad (4)$$

$$\|\hat{\boldsymbol{\theta}}_\rho - \boldsymbol{\theta}^*\|_2 \lesssim \frac{e^{\kappa_E}}{\lambda_2} \left( \sqrt{\frac{n_{\max}(n+r)}{L}} + \rho\kappa\sqrt{n} \right) \quad (5)$$

*where $\lambda_2 = \lambda_2(\mathcal{L}_{\mathbf{A}})$, $r := \kappa_E + \log \kappa$ provided that $L \leq n^8 e^{5\kappa_E} \max\{1, \kappa\}$, and $L$ is large enough so that the right hand side of Equation (4) is smaller than a sufficiently small constant $C > 0$. In particular, if we set*

$\rho = c_\rho/\kappa\sqrt{n_{\max}/L}$ *for some* $c_\rho > 0$, *then*

$$\|\hat{\boldsymbol{\theta}}_\rho - \boldsymbol{\theta}^*\|_\infty \lesssim \frac{e^{2\kappa_E}}{\lambda_2} \frac{n_{\max}}{n_{\min}} \sqrt{\frac{n+r}{L}} + \frac{e^{\kappa_E}}{\lambda_2} \sqrt{\frac{n_{\max}(\log n + r)}{L}},$$

$$\|\hat{\boldsymbol{\theta}}_\rho - \boldsymbol{\theta}^*\|_2 \lesssim \frac{e^{\kappa_E}}{\lambda_2} \sqrt{\frac{n_{\max}(n+r)}{L}}. \qquad (6)$$

As a brief sketch, the proof is based on a gradient descent procedure initialized at $\boldsymbol{\theta}^{(0)} = \boldsymbol{\theta}^*$ and the idea is to control $\|\boldsymbol{\theta}^{(T)} - \hat{\boldsymbol{\theta}}_\rho\|_\infty$ using the linear convergence property and $\|\boldsymbol{\theta}^{(T)} - \boldsymbol{\theta}^*\|_\infty$ using the leave-one-out technique in Chen et al. (2019) and Chen et al. (2020). In fact, our work confirms that such a line of argument extends to more general graph topologies beyond the Erdös-Rényi graph, which is non-trivial. The proof details can be found in Appendix A.2.

**Interpretation of key terms**

The upper bound in Equation (4) contains several distinct terms, which interact with each other in non-trivial ways and express different aspects of the intrinsic difficulty of the estimation task.

- The factor $\frac{e^{\kappa_E}}{\lambda_2(\mathcal{L}_\mathbf{A})}$ combines two sources of statistical hardness: the *maximal gap* in performance $\kappa_E$ among the ranked items over the edge set $E$, and the *algebraic connectivity* $\lambda_2(\mathcal{L}_\mathbf{A})$ of the comparison graph. It is intuitively clear that the larger the performance gap among the compared items, the more difficult it is to accurately estimate the model parameters. Furthermore, the smaller the algebraic connectivity, the less connected the comparison graph is, due to the presence of bottlenecks[1]. This in turn will increase the chance of obtaining a highly erroneous ranking or of gathering data from which a global ranking cannot be elicited at all. The minimal and maximal degrees $n_{\min}$ and $n_{\max}$ further quantify the impact of the connectivity of the comparison graph.

- We note that the factor $\frac{1}{\lambda_2(\mathcal{L}_\mathbf{A})}$ can be equivalently replaced with $\frac{1}{\lambda_2(\mathcal{I})}$ (see Lemma 2 in Appendix A.2). Here, $\mathcal{I} := \nabla^2 \ell_0(\boldsymbol{\theta}^*; \mathbf{y})$ is the Fisher information matrix at $\boldsymbol{\theta}^*$ and $\lambda_2(\mathcal{I})$ its smallest non-zero eigenvalue. The fact that the bound depends on the Fisher information is not too surprising. This is so, since this quantity in exponential families quantifies the curvature of the likelihood and the intrinsic difficulty of estimating $\boldsymbol{\theta}^*$.

- Our bounds depend on both $\kappa$ and $\kappa_E$, which is non-standard in the literature. By definition, $\kappa_E \leq \kappa$ and in many cases, $\kappa_E$ can be much smaller than $\kappa$. We discuss this further in Section 5.

- The term $r := \kappa_E + \log \kappa$ shows the impact of large $\kappa$ and $\kappa_E$. When $\kappa \lesssim n$ and $\kappa_E \lesssim \log n$, $r$ is negligible.

---

[1]Here, bottlenecks can be formally described as small connected subgraphs with very few edges separating dense portions of the graph.

We will consider this parameter range throughout the paper unless stated otherwise.

- The term $\sqrt{\frac{n}{L}}$ describes explicitly the impact of a high-dimensional parameter space on the estimation problem in relation to $L$, the number of samples for each comparison, which can be thought of as a measure of the sample size required for each of the $n$ parameters. The inverse root dependence on $L$ is to be expected and, we conjecture, not improvable.

*Remark* 1. In the case of dense graphs, e.g., complete graphs, $\lambda_2(\mathcal{L}_\mathbf{A})$ is large enough so that even $L = 1$ will ensure a consistent estimator as $n \to \infty$. But for sparse graphs, $L$ needs to be larger to compensate for weaker connectivity. The assumption that $L \leq n^8 e^{5\kappa_E} \max\{1, \kappa\}$ is a technical condition. There is nothing special in the exponent for $n$. Any fixed number larger than 8 can be used which will only affect the constants in the bounds. The condition $L \leq n^8 e^{5\kappa_E} \max\{1, \kappa\}$ may seem counter-intuitive, since it places an upper bound on the sample size. But a control over $L$ is needed because as $L$ gets larger, the optimal choice of the regularization parameter $\rho = c_\rho \frac{1}{\kappa} \sqrt{\frac{n_{\max}}{L}}$ gets smaller and, accordingly, the convergence rate of the gradient descent procedure upon which our proof is based degrades. The optimal choice $\rho = c_\rho \frac{1}{\kappa} \sqrt{\frac{n_{\max}}{L}}$ depends on $\kappa$, which is unknown before an estimator is produced, however, one can set $\rho = c_\rho \sqrt{\frac{n_{\max}}{L}}$ and the upper bound will only change by a factor $\max\{1, \kappa\}$ in the first term of Equation (6).

## 2.1 COMPARISON TO OTHER WORK

To the best of our knowledge, Yan et al. (2012); Hajek et al. (2014); Shah et al. (2016); Negahban et al. (2017); Agarwal et al. (2018); Hendrickx et al. (2019, 2020) are the only existing papers that study estimation error for the BTL model on a comparison graph with general topology. Since Negahban et al. (2017); Agarwal et al. (2018); Hendrickx et al. (2019, 2020) estimate the the preference scores $\mathbf{w}^*$ rather than $\boldsymbol{\theta}^*$, we cannot directly compare our results with theirs because there is no tight two-sided relationship between their metrics of error and ours. Therefore, here we only compare our results to those in Yan et al. (2012); Hajek et al. (2014); Shah et al. (2016), as is summarized in Table 1. We include the comparison to the other four papers in Appendix A.1.

$\ell_\infty$ **loss:** Yan et al. (2012) establish an $\ell_\infty$-bound depending on $n_{ij}$, the number of common neighbors of item $i$ and item $j$ in the comparison graph, under a strong assumption that $n_{ij} \geq cn$ for some constant $c \in (0, 1)$. This constraint on graph topology is stronger than ours since it requires the graph to be dense. In particular, when the comparison graph comes from an Erdös-Rényi model $ER(n, p)$, $\min_{i,j} n_{ij} \asymp np^2$. Then the conditions in Yan et al. (2012) requires $p$ to be bounded away from 0 and their bound becomes $\frac{e^\kappa}{p} \sqrt{\frac{\log n}{npL}}$, while our bound is $\frac{e^{2\kappa_E}}{\sqrt{p}} \sqrt{\frac{\log n}{npL}}$. Our

bound is tighter for moderate or small $\kappa_E$, and importantly, allows $p$ to vanish. Furthermore, in Section 5, we show by some specific examples that $\min_{i,j} n_{ij}$ could be 0 even for many fairly dense graphs, to illustrate that the Yan et al. (2012) upper bound cannot apply to many realistic settings.

$\ell_2$ **loss:** Hajek et al. (2014); Shah et al. (2016) consider constrained MLE $\hat{\boldsymbol{\theta}} := \min_{\|\boldsymbol{\theta}\|_\infty \leq B} \ell_0(\boldsymbol{\theta})$ for a known parameter $B$ such that $\|\boldsymbol{\theta}^*\|_\infty \leq B$. Setting aside the fact that their results require stricter conditions than ours, our $\ell_2$ bound is tighter than theirs for general parameter settings with moderate $B, \kappa$ and for a broad range of graphs with moderate $\lambda_2(\mathcal{L}_\mathbf{A})$, i.e., not too sparse or irregular.

| Norm | Reference | Upper bound |
|------|-----------|-------------|
| $\|\cdot\|_\infty$ | Yan et al. (2012) | $\frac{e^\kappa}{\min_{i,j} n_{ij}} \sqrt{\frac{n_{\max} \log n}{L}}$ |
| | **Our work** | See Theorem 1 |
| $\|\cdot\|_2^2$ | Hajek et al. (2014) | $e^{8B} \frac{|E| \log n}{\lambda_2(\mathcal{L}_A)^2 L}$ |
| | Shah et al. (2016) | $e^{8B} \frac{n \log n}{\lambda_2(\mathcal{L}_A) L}$ |
| | **Our work** | $\frac{e^{2\kappa_E}}{\lambda_2(\mathcal{L}_\mathbf{A})^2} \frac{n_{\max} n}{L}$ |

**Table 1:** Comparison of results in literature.

We re-emphasize that Hendrickx et al. (2020) also provide upper bounds for a general fixed comparison graph that matches an instance-wise lower bound, for their parameter of interest $\mathbf{w}^* := (e^{\theta_1^*}, \ldots, e^{\theta_n^*})^\top$, instead of $\boldsymbol{\theta}^*$. However, their error metric, i.e., $\sin(\hat{\mathbf{w}}, \mathbf{w}^*)$, is quite different from other similar papers in the BTL literature, including our work. As such, it is not clear how to compare to their results. Furthermore, as noted in Section 1, from the perspective of ranking, an entry-wise metric like $\|\cdot\|_\infty$ is more informative than vector-level metrics like $\|\cdot\|_2$ and $\sin(\cdot, \cdot)$.

## 2.2 SPECIAL CASES OF GRAPH TOPOLOGIES

We can check some common types of comparison graph topologies and see in what order the necessary sample complexity $N_{\text{comp}} = |E|L$ needs to be to achieve consistency, i.e., $\|\hat{\boldsymbol{\theta}} - \boldsymbol{\theta}^*\|_\infty = o(1)$. The results are summarized in Table 2. For path and star graphs, we used the specialized bounds in Propositions 3 and 4. As shown in Table 2, our

| Graph | $N_{\text{comp}}$ (Yan et al., 2012) | $N_{\text{comp}}$ **(Our work)** |
|-------|------------|------------|
| **Complete** | $\Omega(n^2)$ | $\Omega(n^2)$ |
| **Bipartite** | N/A | $\Omega(n^2)$ |
| **Path** | N/A | $\omega(e^{2\kappa_E} n^2 \log n)$ |
| **Star** | N/A | $\omega(e^{2\kappa_E} n \log n)$ |
| **Barbell** | N/A | $\omega(e^{2\kappa_E} n^5 \log n)$ |

**Table 2:** Magnitude of $N_{\text{comp}}$ to ensure $\|\hat{\boldsymbol{\theta}} - \boldsymbol{\theta}^*\|_\infty = o(1)$.

bound now applies to a much broader class of graph topolo-

gies under the $\ell_\infty$-norm compared to Yan et al. (2012).

*Remark* 2. For the path graph, star graph, and barbell graph, the necessary sample complexity induced by directly applying our $\ell_\infty$ bound is larger than the sample complexity induced by the $\ell_2$ bound in Shah et al. (2016), though they require more stringent conditions than ours. Thus we provide specialized sharp upper bounds in the case of path and star graph in Proposition 3 and 4. Additionally, in Section 4, we illustrate that by applying a *unique* sub-additivity property of $\ell_\infty$-loss, we can achieve a much smaller sample complexity in graphs with bottlenecks like the barbell graph.

**Erdös-Rényi graph:** By applying a union bound on $\lambda_2(\mathcal{L}_\mathbf{A})$, $n_{\max}$, and $n_{\min}$ to the sample-wise bounds in Theorem 1, we obtain a corollary in the setting where the comparison graph follows the Erdös-Rényi model $ER(n, p)$.

**Corollary 2** (Erdös-Rényi graph)**.** *As a corollary to Theorem 1, suppose that the comparison graph comes from an Erdös-Rényi graph $ER(n, p)$, then under the same conditions, with probability at least $1 - O(n^{-4})$, it holds that*

$$\|\hat{\boldsymbol{\theta}}_\rho - \boldsymbol{\theta}^*\|_\infty \lesssim e^{2\kappa_E} \sqrt{\frac{\log n}{np^2 L}}, \|\hat{\boldsymbol{\theta}}_\rho - \boldsymbol{\theta}^*\|_2 \lesssim e^{\kappa_E} \sqrt{\frac{1}{pL}}.$$

The full form of Corollary 2 with a proof can be found at the end of Appendix A.2. For the Erdös-Rényi comparison graph $ER(n, p)$, the tightest $\ell_\infty$-norm error bound $e^{2\kappa} \sqrt{\frac{\log n}{npL}}$ is proved in Chen et al. (2019) and Chen et al. (2020). Han et al. (2020) establish an $\ell_\infty$-norm upper bound of $e^{2\kappa} \sqrt{\frac{\log n}{np}} \cdot \frac{\log n}{\log(np)}$. Negahban et al. (2017) obtain an $\ell_2$-norm upper bound of $e^{4\kappa} \frac{\log n}{pL}$ and a lower bound of $e^{-\kappa} \frac{1}{pL}$. Thus the derived $\ell_2$-bound in Corollary 2 in Erdös-Rényi case is minimax optimal.

In this case our derived $\ell_\infty$-bound cannot achieve the rate established in Chen et al. (2019), Chen et al. (2020), though our $\ell_2$-bound exhibits the optimal rate proved in Negahban et al. (2017). The reason why our bound does not imply the optimal $\ell_\infty$-rate under a Erdös-Rényi comparison graph is that our bound is a sample-wise bound and thus cannot leverage some regular property of Erdös-Rényi graph beyond algebraic connectivity and degree homogeneity that is exhibited with high probability.

**Tree graphs:** For extremely sparse graphs like tree graphs, the general upper bound in Theorem 1 is loose compared to the lower bound in Theorem 5. Therefore, we separately prove some sharp upper bounds for path and star graphs as a complement to our general theory, in these frequently studied cases. For example, single-elimination sports tournaments are commonly designed as a binary tree graph. By the spectral property of path and star graphs (see Appendix A.6), one can verify that the upper bounds in both norms match the $\ell_\infty$ lower bound in Theorem 5 and the $\ell_2$ lower bound in Shah et al. (2016), up to $\sqrt{\log n}$ and $e^{2\kappa_E}$ factors.

**Proposition 3** (Path graph). *Suppose the comparison graph is a path graph $([n], E)$ with $E = \{(i, i+1)\}_{i \in [n-1]}$ and $L > c e^{2\kappa_E} n \log n$ for some universal constant $c$, then with probability at least $1 - n^{-4}$, the vanilla MLE $\hat{\boldsymbol{\theta}}_0$ satisfies*

$$\|\hat{\boldsymbol{\theta}}_0 - \boldsymbol{\theta}^*\|_\infty \lesssim e^{\kappa_E} \sqrt{\frac{n \log n}{L}},$$

$$\|\hat{\boldsymbol{\theta}}_0 - \boldsymbol{\theta}^*\|_2 \lesssim e^{\kappa_E} n \sqrt{\frac{\log n}{L}}.$$

**Proposition 4** (General tree graph). *Suppose the graph is a tree graph $([n], E)$ where each item $i$ and $j$ are compared $L$ times such that $L > c e^{2\kappa_E} n \log n$ for some universal constant $c$. Then with probability at least $1 - n^{-4}$, the vanilla MLE $\hat{\boldsymbol{\theta}}_0$ satisfies*

$$\|\hat{\boldsymbol{\theta}}_0 - \boldsymbol{\theta}^*\|_\infty \lesssim e^{\kappa_E} \sqrt{\frac{D \log n}{L}},$$

$$\|\hat{\boldsymbol{\theta}}_0 - \boldsymbol{\theta}^*\|_2 \lesssim e^{\kappa_E} \sqrt{\frac{D n \log n}{L}},$$

*where $D := \max_{i,j} |\text{path}(i,j)|$ is the diameter. In particular, for star graph, the upper bound is given by $D = 1$.*

The full form of Proposition 3 and Proposition 4 with proofs are found in Appendix A.2. Briefly, the proofs leverage the closed-form solution of vanilla MLE under the tree graph.

## 3 LOWER BOUNDS

In this section, we derive a minimax lower bound for the $\ell_\infty$ loss. Towards that end, we first introduce some new notation. Let $N_{\text{comp}}$ be the total number of comparisons that have been observed, so in our setting, $N_{\text{comp}} = |E|L$ where $|E|$ is number of edges in the comparison graph $\mathcal{G}$. Denote the two items involved in the $i$-th comparison as $(i_1, i_2)$ such that $i_1 < i_2$. Let $\tilde{\mathcal{L}}_A = \frac{1}{N_{\text{comp}}} \sum_{i=1}^{N_{\text{comp}}} (\mathbf{e}_{i_1} - \mathbf{e}_{i_2})(\mathbf{e}_{i_1} - \mathbf{e}_{i_2})^\top$ be the normalized graph Laplacian with pseudo inverse $\tilde{\mathcal{L}}_A^\dagger$ and eigenvalues $0 = \lambda_1(\tilde{\mathcal{L}}_A) \leq \lambda_2(\tilde{\mathcal{L}}_A) \leq \cdots \leq \lambda_n(\tilde{\mathcal{L}}_A)$. With the main notation in place, our minimax lower bound is summarized in the following result.

**Theorem 5.** *Assume that the comparison graph $\mathcal{G}$ is connected and the sample size $N_{\text{comp}} \geq \frac{c_2 \text{tr}(\tilde{\mathcal{L}}_A^\dagger)}{e^{2\kappa} \kappa^2}$, any estimator $\widetilde{\boldsymbol{\theta}}$ based on $N_{\text{comp}}$ comparisons with outcomes from the BTL model satisfies*

$$\sup_{\boldsymbol{\theta}^* \in \Theta_\kappa} \mathbb{E}\left[\|\widetilde{\boldsymbol{\theta}} - \boldsymbol{\theta}^*\|_\infty^2\right] \gtrsim \frac{e^{-2\kappa}}{n N_{\text{comp}}} \times$$

$$\max\left\{n^2, \max_{n' \in \{2,\ldots,n\}} \sum_{i=\lceil 0.99 n' \rceil}^{n'} [\lambda_i(\tilde{\mathcal{L}}_A)]^{-1}\right\}$$

*where $\Theta_\kappa = \{\theta \in \mathbb{R}^n : \mathbf{1}_n^\top \boldsymbol{\theta} = 0, \|\boldsymbol{\theta}\|_\infty \leq \kappa\}$.*

The proof of Theorem 5 largely leverages the lower bound construction from Theorem 2 in Shah et al. (2016). The main modification in adapting it to our setting is to construct an $\ell_\infty$-packing set. This is done by utilizing the *tight* topological equivalence of $\ell_\infty$ and $\ell_2$ norms in finite dimensions.

We can compare this lower bound with the upper bound in Theorem 1. In our setting, the comparisons distribute evenly over all pairs, so $N_{\text{comp}} = |E|L$, and $\lambda_i(\tilde{\mathcal{L}}_A) = \frac{1}{|E|}\lambda_i(\mathcal{L}_A)$. Thus, given a comparison graph with $\lambda_2(\tilde{\mathcal{L}}_A) \asymp \frac{1}{n}$, the lower bound becomes

$$\sup_{\boldsymbol{\theta}^* \in \Theta_\kappa} \mathbb{E}\left[\|\widetilde{\boldsymbol{\theta}} - \boldsymbol{\theta}^*\|_\infty\right] \gtrsim e^{-\kappa} \sqrt{\frac{n}{N_{\text{comp}}}} \qquad (7)$$

In $ER(n, p)$ case, this lower bound becomes $e^{-\kappa} \sqrt{\frac{1}{npL}}$ which matches the upper bound in Chen et al. (2019). For some "regular" graph topology with $\lambda_2(\tilde{\mathcal{L}}_A) \asymp \frac{1}{n}$ like complete graph, expander graph with $\phi = \Omega(n)$ and complete bipartite graph with two partition sets of size $\Omega(n)$, the upper bound becomes

$$\|\hat{\boldsymbol{\theta}}_\rho - \boldsymbol{\theta}^*\|_\infty \lesssim e^{2\kappa} \sqrt{\frac{n \log n}{N_{\text{comp}}}}.$$

Therefore, when the comparison graph topology is sufficiently regular, our upper bound matches the lower bound up to a $\log n$ factor and a factor of $e^{3\kappa}$. As a final remark, Negahban et al. (2017) show that the minimax lower bound for $\ell_2$-loss and Erdös-Rényi comparison graph $ER(n, p)$ is $e^{-\kappa} \frac{1}{pL}$, which matches our $\ell_2$ upper bound up to a factor of $e^{2\kappa}$.

## 4 IMPLICATIONS FOR TOURNAMENT DESIGN

In this section, we discuss how our results can be leveraged to construct more efficient tournament design from a ranking perspective in sports leagues.

As discussed in Section 2.2, for some comparison graphs with small $\lambda_2(\mathcal{L}_{\mathbf{A}})$, the requirement on $L$ and $N_{\text{comp}}$ for consistency is stringent. However, as we show next, we can significantly relax the requirement on the sample complexity $N_{\text{comp}}$ by adaptively varying the number pairwise comparisons observed over different subsets of the items in a manner that leverages different degrees of connectivity of the comparison graphs.

The basic idea is that model parameters corresponding to a subset of items inducing a highly connected sub-graph require relatively few observations. On the other hand, the outcomes of comparisons with items corresponding to nodes of the comparison graph that are part of a "graph bottleneck" are especially important in yielding accurate global ranking and, therefore, should be more heavily sampled (in the

sense of having a larger number $L$ of observations). The case of a Barbell graph consisting of two complete sub-graphs connected by few "bridge" edges (as is shown in Figure 2) is an extreme illustration of this situation and will be discussed below. In this case, it is clear that the parameters corresponding the items adjacent to the bridge edges ought to be estimated with higher accuracy and therefore, for those items $L$ should be set larger. Furthermore, it is possible to estimate the model parameters separately over different sub-graphs and combine these estimators in a way that could lead to an improved rate, compared to a joint or omnibus estimator. Indeed, the next result shows that the $\ell_\infty$-error rate of the combined estimator is bounded by the sum of the error rates for estimating the parameters of the individual sub-graphs.

Formally, let $I_1, I_2, I_3$ be three subsets of $[n]$ such that $\cup_{j=1}^3 I_j = [n]$ and, for each $j \neq k$, $I_j \not\subseteq I_k$ and for $i = 1, 2$, $I_i \cap I_3 \neq \emptyset$. Assume that the sub-graphs induced by $I_j$'s are connected and the number of comparisons for all pairs can be different across sub-graphs. Let $\boldsymbol{\theta}^*$ be the vector of preference scores in the BTL model over $n$ items and $\hat{\boldsymbol{\theta}}_{(j)}$ be the MLE of $\boldsymbol{\theta}^*_{(j)} \in \mathbb{R}^{|I_j|}$ for the BTL model involving only items in $I_j$, $j = 1, 2, 3$. Also define the augmented version $\tilde{\boldsymbol{\theta}}_{(j)} \in \mathbb{R}^n$ such that $\tilde{\boldsymbol{\theta}}_{(j)}(I_j) = \hat{\boldsymbol{\theta}}_{(j)}$.

Now take two nodes $t_1 \in I_1 \cap I_3$, $t_2 \in I_2 \cap I_3$, and let $\delta_3 = \tilde{\boldsymbol{\theta}}_{(1)}(t_1) - \tilde{\boldsymbol{\theta}}_{(3)}(t_1)$, $\delta_2 = \tilde{\boldsymbol{\theta}}_{(3)}(t_2) - \tilde{\boldsymbol{\theta}}_{(2)}(t_2)$. An ensemble estimator *add-MLE* $\hat{\boldsymbol{\theta}} \in \mathbb{R}^n$ is a vector such that $\hat{\boldsymbol{\theta}}(I_1) = \hat{\boldsymbol{\theta}}_{(1)}$, $\hat{\boldsymbol{\theta}}(S_2) = \hat{\boldsymbol{\theta}}_{(2)}(S_2) + \delta_3 + \delta_2$, and $\hat{\boldsymbol{\theta}}(S_3) = \tilde{\boldsymbol{\theta}}_{(3)}(S_3) + \delta_3$, where $S_2 = I_2 \setminus I_1$ and $S_3 = I_3 \setminus (I_1 \cup I_2)$. Notice that the value of $\hat{\boldsymbol{\theta}}$ depends on the choice of $t_1, t_2$, but the estimation error of all ensemble estimators can be well-bounded, as is shown in Proposition 6.

**Proposition 6** (Subadditivity of $\ell_\infty$-loss in BTL). *Under the setting above, for any add-MLE $\hat{\boldsymbol{\theta}} \in \mathbb{R}^n$ based on $\hat{\boldsymbol{\theta}}_{(1)}, \hat{\boldsymbol{\theta}}_{(2)}, \hat{\boldsymbol{\theta}}_{(3)}$, it holds that*

$$d_\infty(\hat{\boldsymbol{\theta}}, \boldsymbol{\theta}^*) \leq 4 \sum_{i=1}^3 d_\infty(\hat{\boldsymbol{\theta}}_{(i)}, \boldsymbol{\theta}^*_{(i)}), \qquad (8)$$

*where $d_\infty(\mathbf{v}_1, \mathbf{v}_2) := \|(\mathbf{v}_1 - \mathrm{avg}(\mathbf{v}_1)\mathbf{1}) - (\mathbf{v}_2 - \mathrm{avg}(\mathbf{v}_2)\mathbf{1})\|_\infty$ and $\mathrm{avg}(\mathbf{x}) := \frac{1}{n}\mathbf{1}_n^\top \mathbf{x}$ for $\mathbf{x} \in \mathbb{R}^n$.*

The proof of Proposition 6 is found in Appendix A.5. For some types of graph topologies the above result can be used to devise a *divide-and-conquer strategy* for estimating the model parameters with better sample complexity than that of an omnibus estimator, i.e., the joint-MLE in our setting. Indeed, as discussed in Section 2.2, for a barbell graph containing two size-$n/2$ complete sub-graphs connected by a single edge, we need $N_{\mathrm{comp}} = \Omega(n^5 \log n)$ for an $o(1)$ error bound of the joint-MLE. From a practical perspective, we note that such a divide and conquer strategy gives flexibility in the number of comparisons in each sub-graph. For example, if we set $L = 1$ for the two complete sub-graphs to get

MLEs $\hat{\boldsymbol{\theta}}_1$, $\hat{\boldsymbol{\theta}}_2$, and set $L = n$ for the two items linking the two sub-graphs to get an MLE $\hat{\boldsymbol{\theta}}_3$, and combine them by shifting $\hat{\boldsymbol{\theta}}_2$ by the difference of two entries of $\hat{\boldsymbol{\theta}}_3$, then a total sample complexity *reduction* to $N_{\mathrm{comp}} = \Omega(n^2)$ will ensure $\ell_\infty$-norm error of order $O(e^{2\kappa_E}\sqrt{\log n}/\sqrt{n}) = o(1)$, because for a complete graph of size $m$, the $\ell_\infty$-norm error is $O(e^{2\kappa_E}\sqrt{\log m}/\sqrt{mL})$. In Example 7 and Appendix A.4.1, we show some simulation results illustrating the advantage of using subadditivity in estimation, where we generalize the add-MLE to Island graph and Barbell graph with multiple bridge edges that can have more than 3 dense sub-graphs.

Note that such flexible tournament design is similar to the idea of *active ranking* (Heckel et al., 2019; Ren et al., 2019), but there is still a substantial difference between our setting and active ranking. Active ranking assumes that one can design the tournament in an *online* manner, so that the next pair of items to be compared is determined by the newest outcomes of comparisons. However, in practice many tournaments can only be designed *offline*, i.e., before any games are played. Under this common setting, our $\ell_\infty$-subadditivity property provides a useful offline approach to efficient tournament design.

## 5 EXAMPLES AND SIMULATIONS

In this section, we conduct numerical experiments on simulated data with two main goals. First, we illustrate the utility of the subadditivity property in Proposition 6 in the case of Island graphs (see Example 7). Second, we demonstrate the relative tightness of our $\ell_\infty$ upper bound compared to Yan et al. (2012), since their work is closest in spirit to ours. Specifically, we compare the two bounds in a setting where analytical comparison is not directly feasible (see Example 8). All of our reproducible code is openly accessible[2].

In the BTL model, the maximal winning probability is $p_{\max}(\kappa) = 1/(1 + e^{-\kappa})$. To get a sense, $p_{\max}(2.20) = 0.900$, $p_{\max}(4.59) = 0.990$. A winning probability larger than 0.99 is fairly rare in practice, so it would not be too constraining to set $\kappa = 2.2$ in our simulation. But analytically our result allows $\kappa$ to diverge with $n$.

In our experiments, we set $\theta_i^* = \theta_1^* + (i-1)\delta$ for $i > 1$ with $\delta = \kappa/(n-1)$. We additionally assign $\theta_1^*$ to ensure that $\mathbf{1}_n^\top \boldsymbol{\theta}^* = 0$, for parameter identifiability. Under this setting, for some special graphs, e.g., the Island graph in Example 7, $\kappa_E$ can be much smaller than $\kappa$, showing an advantage of our upper bound in representing the dependency on the maximal performance gap $\kappa_E$ along the edge set, rather than $\kappa$ the whole vertex set. However, there may be some cases where the majority of edges have small performance gaps and only a few edges have large gaps. Here, the control in the upper bound purely by $\kappa_E$ can again be loose. An interesting future

---

[2]Repo: `https://github.com/MountLee/btl_mle_l_inf`

direction is to make upper bounds tighter in such cases by including more structural parameters, like the proportion of small-gap edges. We include some illustrative examples in Appendix A.4.

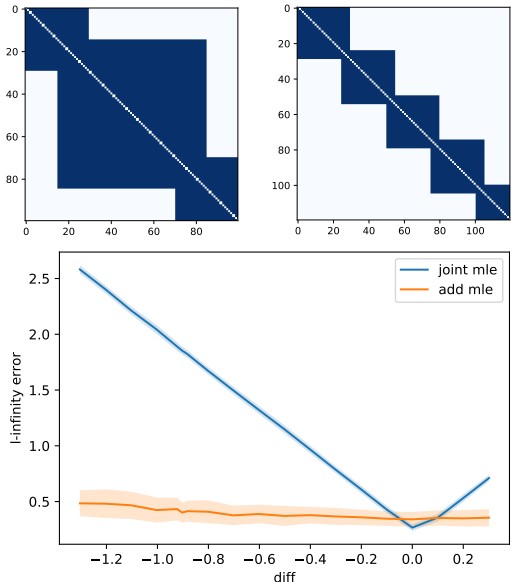

**Figure 1:** Left: Adjacency matrix of a 3-Island graph, with yellow indicating 1 and purple indicating 0; $\lambda_2(\mathcal{L}_{\mathbf{A}}) = 11.92$. Right: Adjacency matrix of a general Island graph, with $n_{\text{island}} = 30$, $n_{\text{overlap}} = 5$, $n = 120$; $\lambda_2(\mathcal{L}_{\mathbf{A}}) = 1.19$. Bottom: comparison of the error of the joint-MLE and the add-MLE. The curve is obtained as the average of 100 trials with one standard deviation shown by the colored area.

**Example 7** (Graph with $\min_{i,j} n_{ij} = 0$). In this case, we intend to illustrate that $\min_{i,j} n_{ij}$ could be 0 or quite close to 0 for even fairly dense graphs, making the upper bound in Yan et al. (2012) less effective. Consider a *3-Island* comparison graph $\mathcal{G}$ with $n$ nodes. The induced sub-graphs on node sets $V_1, V_2, V_3$ with $|V_i| = n_i$ are complete graphs, where $V_1 \cap V_3 = \emptyset$, $V_1 \cup V_2 \cup V_3 = [n]$, and $V_i \cap V_2 \neq \emptyset$ for $i = 1, 3$. There is no edge except for those within $V_1, V_2, V_3$. This graph $\mathcal{G}$ is connected, and can be fairly dense if we make $n_2$ large, but $\min_{i,j} n_{ij} = 0$ always holds since $V_1 \cap V_3 = \emptyset$ and the two induced sub-graphs are complete. See Figure 1 left panel for a visualization of the adjacency matrix of such a graph.

We can also consider more general *Island* graphs. A general Island graph is determined by $n$, the size of the graph, $n_{\text{island}}$, the size of island sub-graphs, and $n_{\text{overlap}}$, the number of overlapped nodes between islands. Each island sub-graph is a complete graph, and there is no edge outside islands. For Island graphs, it holds that $\min_{i,j} n_{ij} = 0$ and $\kappa_E \approx \kappa \cdot n_{\text{island}}/n$. Figure 1 top panel shows the adjacency matrix of two Island graphs. Figure 1 bottom panel shows the comparison of the $\ell_\infty$-error of the joint-MLE and the add-MLE (see the detailed definition in Ap-

pendix A.4) while varying the difference in the average of preference scores of each island sub-graph, where we set $n_{\text{island}} = 50, n_{\text{overlap}} = 5, L = 10$. Every point on the lines is the average of 100 trials. It can be seen that the add-MLE by the divide-and-conquer strategy largely dominates the joint-MLE in $\ell_\infty$-error.

In Example 7, we show a common family of graphs which is fairly dense while $\min_{i,j} n_{ij} = 0$, so that the upper bound in Yan et al. (2012) does not hold. Next in Example 8 we consider another family of graphs where their upper bound holds but still relatively looser than our bound.

**Example 8** (Barbell graph with random *bridge* edges). Consider a generalized Barbell graph $\mathcal{G}$ containing $n = n_1 + n_2$ nodes, where the induced sub-graph on nodes $\{1, \cdots, n_1\}$ and $\{n_1 + 1, \cdots, n\}$ are complete graphs, and the two sub-graphs are connected by some bridge edges $(i, j)$ for some $1 \leq i \leq n_1$ and $n_1 + 1 \leq j \leq n$. Denote the set of bridge edges as $E_l$, then $|E_l|/(n_1 n_2)$ quantifies the connectivity of $\mathcal{G}$: the larger $|E_l|/(n_1 n_2)$ is, the denser or more regular $\mathcal{G}$ is.

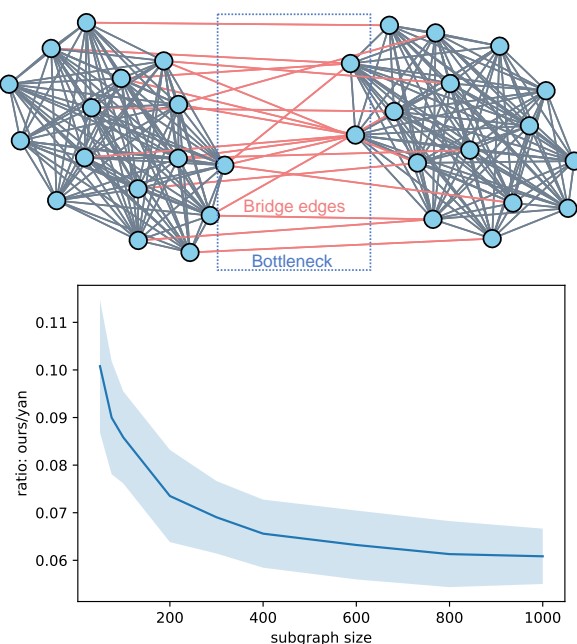

**Figure 2:** Top: visualization of a Barbell graph with random bridge edges. Bottom: The ratio of our bound and the bound in Yan et al. (2012) under the Barbell graph with random bridge edges and sub-graph size $n_1 = n_2 = n_s$ varying. The curve is obtained as the average of 100 trials with one standard deviation shown by the colored area.

In Figure 2 we show a comparison of the real $\ell_\infty$-loss $\|\hat{\boldsymbol{\theta}} - \boldsymbol{\theta}^*\|_\infty$, and the upper bounds of $\ell_\infty$-error in Yan et al. (2012) and our paper. We include this relative comparison to numerically demonstrate that our bound is in general tighter than Yan et al. (2012), since there is no *known* analyt-

ical relationship between $\min_{i,j} n_{ij}$ and $\lambda_2(\mathcal{L}_A)$ for general graphs. In our experiment, we set $n_1 = n_2 = n_s$, $L = 10$, and randomly link $|E_l| = n_1 n_2 p$ edges between the two complete sub-graphs, and vary $n_s$ from 50 to 1000 with $p = 3 \log(n_s)/n_s$. Every point on the line is the average of 100 trials. It can be seen that our upper bound has a faster vanishing rate, compared to Yan et al. (2012) for this simulated scenario. This is evident as the plotted ratio of our upper bound relative to the upper bound in Yan et al. (2012) has a steady decreasing trend, as $n$ increases. It should be noted that there are leading constant factors in both upper bounds, and for convenience we set them to be 1 for both bounds. Thus, one should focus on the trend of the curve rather than the magnitude of the ratio in Figure 2.

## 6 DISCUSSION

In this work we provide a sharp risk analysis of the MLE for the BTL global ranking model, under a more general graph topology, in the $\ell_\infty$-loss. This addresses a major gap in the BTL literature, in extending the comparison graph to more general and thus more practical settings, compared to dense graph setting in Yan et al. (2012). Specifically we derive a novel upper bound for the $\ell_\infty$ and $\ell_2$-loss of the BTL-MLE, showing explicit dependence on the algebraic connectivity of the graph, the sample complexity, and the maximal performance gap between compared items. We also derive lower bounds for the $\ell_\infty$-loss and analyze specific topologies under which the MLE is nearly minimax optimal. We also show that the $\ell_\infty$-loss satisfies a unique subadditivity property for the BTL MLE and utilize our derived bounds for efficient tournament design. We note that our upper bound is suboptimal in the cases where the graph topology is extremely sparse or irregular. Although we provide sharp upper bounds for path and star graphs as separate propositions, we still miss optimality other graph topologies. A good future direction would be to optimize the upper and lower bounds in such comparison graph regimes. Another promising direction is to extend this analysis to the multi-user ranking models as in Jin et al. (2020).

**Acknowledgments**

We would like to thank Heejong Bong from the Carnegie Mellon University (CMU) Department of Statistics & Data Science, for his valuable feedback and discussions during this work. We would also like to thank the anonymous reviewers for their feedback which greatly helped improve our exposition.

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
