# OpenReview forum: "$\ell_{\infty}$-Bounds of the MLE in the BTL Model under General Comparison Graphs"
_auai.org/UAI/2022/Conference — UAI 2022 Poster_

### Official Review · Reviewer_FkM6 · 2022-04-12

**Q2(1) Originality/Novelty:** 3
**Q2(2) Significance/Impact:** 3
**Q2(3) Correctness/Technical Quality:** 3
**Q2(6) Clarity Of Writing:** 4
**Q6 Overall Score:** 7
**Q8 Confidence In Your Score:** 3

**Q1 Summary And Contributions:**

The paper addresses a gap in the BTL model literature, namely to analyze the performance of the MLE in the $l_\infty$ loss under general graph topologies (these represent the pairwise comparisons). Previous work considered either the $l_\infty$ loss (but only for restricted graph classes, such as dense graphs) or general graph topologies, but then not the $l_\infty$ loss.

**Q2 Assessment Of The Paper:**

More detailed information regarding each of these aspects is given below:

**Q2(4) Quality Of Experiments (Optional):**

3: Good: The experimental evaluation is adequate, and the results convincingly support the main claims.

**Q2(5) Reproducibility:**

4: Excellent: Key resources (e.g., proofs, code, data) are available and key details (e.g., proof sketches, experimental setup) are comprehensively described for competent researchers to confidently and easily reproduce the main results.

**Q3 Main Strengths:**

Originality/Novelty:
- The paper addresses an open problem in the BTL model literature and offers highly non-trivial results.
- This problem is dealt with thoroughly and convincingly


Significance/Impact:
- I would assume that the paper has impact mainly in the BTL literature and to some degree applications of the BTL model. (Disclaimer: I am personally not familiar with the rich literature in this area)


Correctness/Technical Quality:
- I did not check the proofs in detail as I'm far from an expert in this topic. In general, I judge the technical quality of this paper to be very high (see more below)


Clarity Of Writing/Reproducibility:
- The quality of writing is excellent. The main message of the paper is presented convincingly and the manuscript in general is very well prepared.

**Q4 Main Weakness:**

Generally, I don't see many weaknesses in this paper. While the scope of the paper could be described as rather narrow and it does not develop completely groundbreaking techniques, for me these are not major weaknesses in this case. On the contrary, I see it as a paper, which convincingly addresses a specific open problem.

**Q5 Detailed Comments To The Authors:**

1. In Sec. 4, the paper introduces a subadditivity property, which suggests that varying L for different item pairs might be useful. As far as I can see, this is only shown practically for the barbell example. I wonder whether some algorithmic approach is possible to find the best choices for L on general graphs.
2. The problem setting is compared in detail to related work in the introduction and in Sec. 2.1. Maybe one could make this even clearer by giving some table in the spirit of the ones in A.1 in the appendix. For example, let the columns be the norms and the rows the graph model. The fields could include the citations.
3. Writing:
- some commata are missing: [page 2] ranking loss, i.e., whether..., [page 2] However, a notable ..., [page 3] In Section 2, we ..., [page 6] In this section, we discuss...
- In Sec. 5 in the text you refer to Fig. 1 left/middle/right panel, but it is two upper and one lower panel.

**Q7 Justification For Your Score:**

In my opinion the paper is a clear accept. It is well-written, on a high technical level and convincingly tackles an open problem in the BTL model literature.

**Q9 Complying With Reviewing Instructions:**

1: Yes.

---

### Official Review · Reviewer_cWWm · 2022-04-12

**Q2(1) Originality/Novelty:** 3
**Q2(2) Significance/Impact:** 3
**Q2(3) Correctness/Technical Quality:** 3
**Q2(6) Clarity Of Writing:** 3
**Q6 Overall Score:** 7
**Q8 Confidence In Your Score:** 1

**Q1 Summary And Contributions:**

This paper furthers Yan's work in extending the comparison graph to more eneral and thus more practical settings.

**Q2 Assessment Of The Paper:**

More detailed information regarding each of these aspects is given below:

**Q2(5) Reproducibility:**

2: Fair: Key resources (e.g., proofs, code, data) are unavailable but key details (e.g., proof sketches, experimental setup) are sufficiently well-described for an expert to confidently reproduce the main results.

**Q3 Main Strengths:**

Fundamentally further Yan's work; gives many contributions to analyzing of the MLE for the BTL model.

**Q4 Main Weakness:**

The organization could be improved.

**Q5 Detailed Comments To The Authors:**

The organization could be improved. Such as "Relevant and related literature" and 2.1 "Comparison to other work" should be merged. The reading flow is not smooth.

I didn't find where Table 1 's results come from.

I also wonder if section 5 is necessary for a theory paper. I prefer more on backgrounds.






**Q7 Justification For Your Score:**

I have no objective for this work. And I am not so familiar with this field. I wish to see other reviewers' comments and to finalize my final judgment.

**Q9 Complying With Reviewing Instructions:**

1: Yes.

---

### Official Review · Reviewer_CEag · 2022-04-14

**Q2(1) Originality/Novelty:** 2
**Q2(2) Significance/Impact:** 2
**Q2(3) Correctness/Technical Quality:** 3
**Q2(6) Clarity Of Writing:** 3
**Q6 Overall Score:** 5
**Q8 Confidence In Your Score:** 2

**Q1 Summary And Contributions:**

In this paper, the authors study the model parameters in the ell_infinity loss of the maximum likelihood estimator in the Bradley-Terry-Luce model and prove various upper and lower bounds.


**Q2 Assessment Of The Paper:**

More detailed information regarding each of these aspects is given below:

**Q2(4) Quality Of Experiments (Optional):**

2: Fair: The experimental evaluation is weak: important baselines are missing, or the results do not adequately support the main claims.

**Q2(5) Reproducibility:**

4: Excellent: Key resources (e.g., proofs, code, data) are available and key details (e.g., proof sketches, experimental setup) are comprehensively described for competent researchers to confidently and easily reproduce the main results.

**Q3 Main Strengths:**

- Rigorous risk analysis of MLE over various graph topologies
- Bounds noticeably improve over Yan et al. [2012]
- Application to tournament design by noting subadditivity of ell_infinity loss in BTL


**Q4 Main Weakness:**

- UAI might not be the right venue for this submission
- Not clear exposition in the introduction


**Q5 Detailed Comments To The Authors:**

- Page 7, right column: If you set \theta_i^* as you do, then how can you still enforce \theta_i^* to sum to zero (which is standard convention in the BTL model)?
- The reader would benefit if you describe the maximum likelihood estimator in the introduction (for example see Section 2 of Yan et al. [2012])

Typos:
- Page 3: “Following the standard in the BTL literature” -> Following the standard assumption in the BTL literature
- Page 3: assume a that —> assume that



**Q7 Justification For Your Score:**

It’s unclear if this submission is appropriate for UAI. For example, I cannot see how the contents of this paper can be presented in a talk/poster at the conference. Maybe a journal in statistics is a better venue? Also, I think the exposition in the introduction must be significantly improved. In the current form, it is not friendly to readers not familiar with MLE.
Post author response, while I'm not totally convinced about the aptness of the submission to UAI, I'm willing to increase my score

**Q9 Complying With Reviewing Instructions:**

1: Yes.

---

### Decision · Program_Chairs · 2022-05-15

**Decision:**

Accept (Poster)

**Comment:**

Meta Review: This paper proposed a \ell_infty norm bound of the maximum likelihood estimator in the BTL Model under general comparison graphs. The result is a good addition to existing \ell_2 norm bounds. While one reviewer still has a reservation about the presentation of the paper, all the reviewers agree that the authors have addressed most of the questions raised by reviewers and the meta-reviewer very well. Thus, I recommend acceptance. Please prepare the camera ready carefully.